# A temperature sensor with a wide spectral range based on a dual-emissive TADF dendrimer system

Changfeng Si[1,3], Tao Wang [1,3], Yan Xu[1], Dongqing Lin[2], Dianming Sun [1] ✉ &
Eli Zysman-Colman [1] ✉

Dual emission from thermally activated delayed fluorescence (TADF) emitters is often difficult to observe, especially in solution, limited by Kasha's rule. Two TADF dendrimers containing N-doped polycyclic aromatic hydrocarbons as acceptors are designed and synthesized. Compound 2GCzBPN, having a strongly twisted geometry, exhibits TADF, while 2GCzBPPZ, possessing a less twisted geometry, shows dual emission associated with the monomer and aggregate that is TADF. The demonstration reveals that 2GCzBPPZ can serve as a temperature sensor with excellent temperature sensitivity and remarkably wide emission color response in solution. By embedding 2GCzBPPZ in paraffin we demonstrate a spatial-temperature sensor that shows a noticeable emission shift from yellow to green and ultimately to blue as the temperature increases from 20 to 200 °C. We finally demonstrate the utility of these TADF dendrimers in solution-processed organic light-emitting diodes.

Thermally activated delayed fluorescence (TADF) materials have attracted much attention in the last decades as alternative emitters for noble-metal-based phosphorescent complexes in organic light-emitting diodes (OLEDs), due to their comparable ability to harvest up to 100% of the triplet excitons to produce light, while simultaneously being less expensive and using more sustainable elements[1,2]. TADF relies on there being a relatively small energy gap ($\Delta E_{ST}$) between the lowest energy triplet state ($T_1$) and the lowest energy singlet state ($S_1$) to enable reverse intersystem crossing (RISC) at ambient temperatures. To obtain a small $\Delta E_{ST}$ in purely organic molecules, the most widely adopted strategy is to introduce a strongly twisted donor-acceptor system. This approach produces an $S_1$ state of intramolecular charge transfer (ICT) character and also results in a compound where there is a small overlap of the electron density between the highest occupied molecular orbital (HOMO) and the lowest unoccupied molecular orbital (LUMO)[3,4]. Most of TADF, and indeed most emissive, compounds only emit from a single, low-lying excited state, adhering to Kasha's rule[5,6].

Several strategies have been advanced to design organic TADF molecules that show dual-emission characteristics from a single component, behavior that could be exploited to produce white-light emission[7], or in bioimaging[8]. Dual emission can be induced where an emissive compound exists in multiple conformations, each with their distinct photophysical properties. First reported by Adachi and co-workers, who designed a donor-acceptor (D-A) TADF emitter, PTZ-TRZ (Supplementary Fig. 1), this compound exists in both a quasi-axial and quasi-equatorial conformation and shows dual ICT fluorescence[9]. Li et al. reported the compound a-DMAc-TRZ, which showed emission from two different conformations[10]. Dual emission could also arise from equilibrated locally excited (LE) and CT excited states[11]. Geng et al. designed two molecules, TMCz-σ-TRZ and DMAC-σ-TRZ, that contain a hexafluoroisopropylidene σ-bridging unit between the donor and acceptor moieties. Both of these compounds showed simultaneous emission from LE and CT states, the latter of which showed TADF character[12]. Ma et al. observed dual emission from MeCz-BP, where the short wavelength emission bands originate from emission from a LE

[1]Organic Semiconductor Centre, EaStCHEM School of Chemistry, University of St Andrews, St Andrews KY16 9ST, UK. [2]State Key Laboratory of Organic Electronics and Information Displays & Institute of Advanced Materials (IAM), Nanjing University of Posts and Telecommunications, Nanjing, China. [3]These authors contributed equally: Changfeng Si, Tao Wang. ✉e-mail: sundianming@hotmail.com; eli.zysman-colman@st-andrews.ac.uk

state from the MeCz donor and the long wavelength emission band was ascribed to CT emission[13]. Another design involves the dual emission from hybrid ICT and intermolecular CT (inter-CT) excited states. Most TADF systems emit from either an ICT excited state or an inter-CT excited state, such as the existence in an exciplex; however, there are a limited number of reports of TADF systems where there is a coexistence of intra- and inter-CT states[14,15]. For instance, Chi and co-workers explored four asymmetrical donor-acceptor-donor' compounds, CPzP, CPzPO, SPzP, and SPzPO (Supplementary Fig. 1), each showing dual emission. In solution, as the concentration increased, the emission maximum of the high-energy ICT bands remained unchanged whereas the low-energy inter-CT bands increased in intensity significantly, which the authors attributed to the enhanced C−H···O hydrogen bonding between adjacent molecules[14]. Besides, some TADF molecules containing an asymmetric triad structure showed dual emission emanating from two different ICT transitions from different donors to a common acceptor. For example, Zhu et al. developed a TADF emitter that combined phenothiazine and N-(1H-indole-5-yl) acetamide as donors, each attached to the same diphenylsulfone acceptor that showed dual emission[8]. The formation of an intermolecular hydrogen-bonded network and the quasi-equatorial conformation of phenothiazine made the dual TADF emission strong both in dilute solution and in the aggregated state.

In addition to the aforementioned strategies, modulating the ratio of excimers/aggregates versus monomer species would be another strategy to obtain dual emission systems[16–22]. Generally, excimer formation is favored when the structure of the emitter is rigid and planar, such as is found in anthracene[23], pyrene[24], and fluorene[25]. As a result, excimers are uncommon in highly twisted donor-acceptor TADF compounds due to steric hindrance that impedes the required intermolecular π-stacking interactions[26,27]. Aggregates are clusters of molecules that are held together by intermolecular interactions and form in the ground state, unlike excimers that form in the excited state[28,29]. However, most emissive aggregates exhibit a single emission band that is red-shifted (J-aggregation) or blue-shifted (H-aggregation) compared to the emission of the monomeric species[30,31]. There are relatively few examples of dual emission TADF systems resulting from a combination of monomer and aggregate emission, especially in the solution state[32].

Optical temperature sensing using organic fluorophores has been developed over the past two decades, there are now examples of organic temperature sensors that show high sensitivity, fast response, simple operation, and have been used in diverse applications such as bioimaging[33], fluorescent thermometers[34], and microfluidics[35]. Detection using these sensors typically relies on the temperature dependence of one of the emission intensity[36], wavelength[37], and lifetime[38]. Most of the small molecule temperature-sensitive fluorescent probes are derived from rhodamine, BODIPY, or molecules emitting from a twisted ICT state, whose emission intensity and/or lifetime are temperature dependent and where the sensors operate typically over a narrow temperature range from around 20 °C to 70 °C[33,39,40]. Organic TADF materials have also shown promise as temperature sensors[41,42]. However, most TADF-based temperature sensors rely only on changes in the emission lifetimes or emission intensity of the materials, which are governed by the endothermic nature of the RISC processes. For example, Farinha and co-workers reported the encapsulation of $C_{70}$ in polymer nanoparticles, this system showed an emission intensity increase with increasing temperature, corresponding to a working range from −75 °C to 105 °C[43]. Borisov and co-workers developed TADF-based temperature probes by encapsulating TADF compounds in a low-oxygen permeability polymer, featuring good temperature sensitivity in the range of 5 °C to 50 °C with 1.4−3.7% $K^{-1}$ change of delayed lifetime, $\tau_d$, at 298 K[42]. However, these two examples of temperature sensors that rely on emission lifetime or intensity change are generally more complex to integrate into devices than conventional temperature sensors due to the need for specialized instrumentation for their excitation and detection, thereby limiting their wider use. A second class of TADF-based temperature sensors relies on changes in emission color; however, there are very few reports of spectral TADF temperature sensors. Hudson and co-workers designed a temperature-responsive polymer by co-polymerization of the TADF monomer, NAI − DMAC, with N-isopropylacrylamide and a blue, fluorescent dopant (tBuODA) as a Förster resonance energy transfer (FRET) acceptor[41]. The polymer was used as a ratiometric temperature sensors that varied its color from red TADF emission at room temperature to blue fluorescence at 70 °C, showing a ratiometric fluorescent thermal response of 32 ± 4% $K^{-1}$ over a temperature range from 20 °C to 70 °C.

In this study, two different rigid and planar N-doped polycyclic aromatic hydrocarbons (PAHs) were employed as the acceptors in two TADF dendrimers containing a second-generation tercarbazole donor dendron, GCz. The chemical structures of the dendrimers 2GCzBPPZ and 2GCzBPN are shown in Fig. 1. Compound 2GCzBPPZ possesses a complex concentration- and temperature-dependent dual emission where the longer wavelength emission band shows TADF. Through a combination of detailed photophysical studies and theoretical calculations, we attribute to the emission from both monomeric and aggregates species. Due to the more twisted geometry and the presence of bulky substituents, 2GCzBPN exhibits TADF and aggregates formation is suppressed. The distinct photophysical properties of

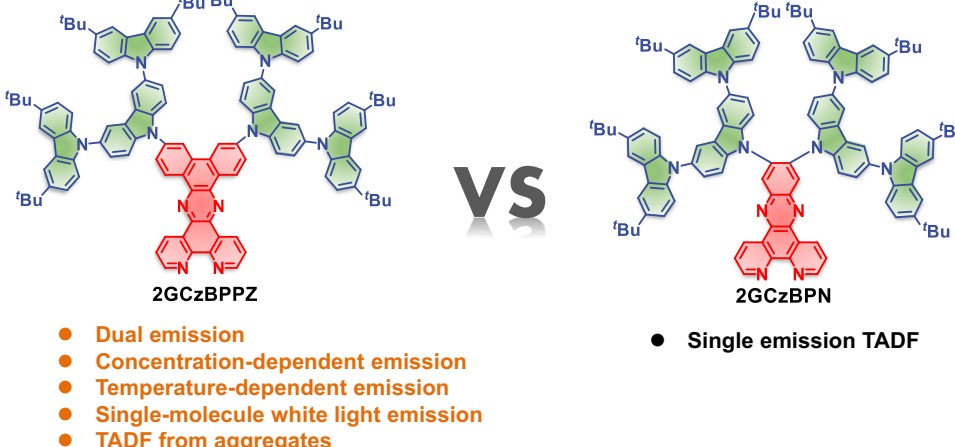

2GCzBPPZ

- Dual emission
- Concentration-dependent emission
- Temperature-dependent emission
- Single-molecule white light emission
- TADF from aggregates

2GCzBPN

- Single emission TADF

**Fig. 1 | Chemical structures of 2GCzBPPZ and 2GCzBPN.** Bullet points present the characteristic photophysical properties of the emitters.

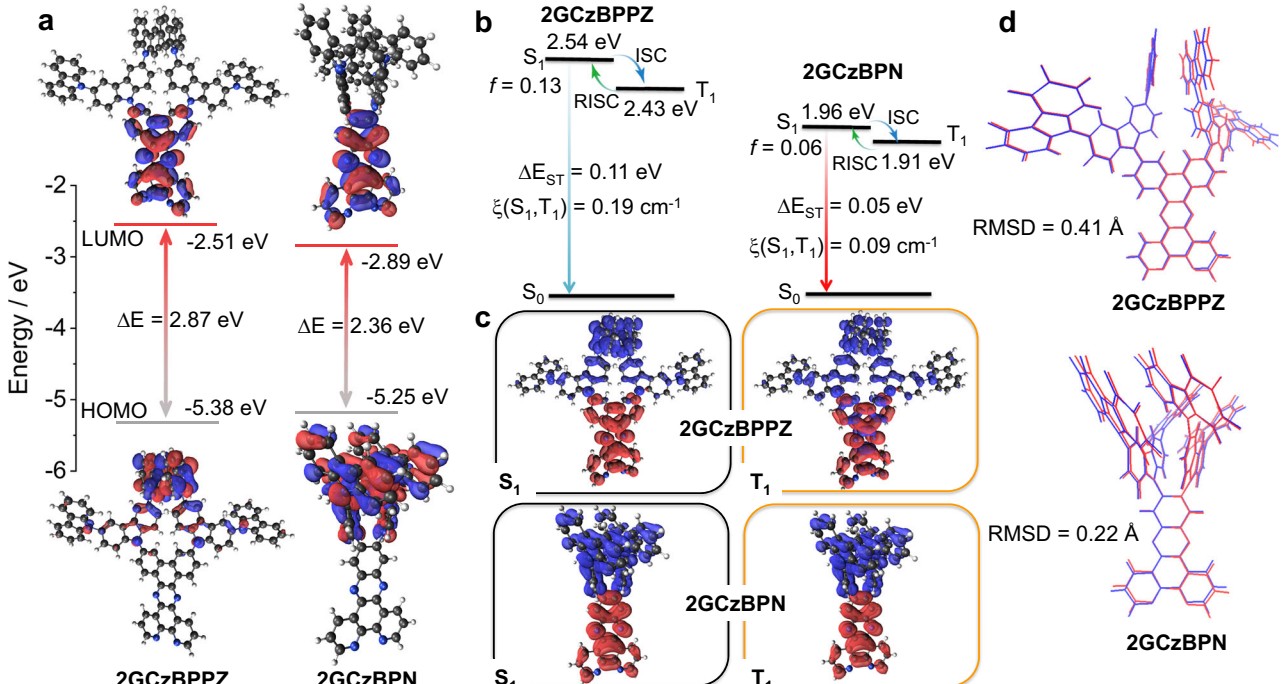

**Fig. 2 | Density-functional theory calculations at the PBE0/6-31 G(d,p) level in the gas phase. a** Frontier molecular orbitals (isovalue: 0.02, ΔE refer to the HOMO-LUMO gap) and (**b**) vertical excitation energy levels of 2GCzBPPZ and 2GCzBPN. ISC, RISC, $\Delta E_{ST}$, $f$, and ξ refer to intersystem crossing, reverse ISC, energy gap between the lowest energy triplet state ($T_1$) and the lowest energy singlet state ($S_1$), oscillator strength and spin–orbit couplingmatrix elements, respectively. **c** Natural transition orbitals of $S_1$ and $T_1$ for 2GCzBPPZ and 2GCzBPN calculated at the optimized $S_0$ geometry, where blue and red represent hole and electron, respectively (isovalue: 0.02). **d** Comparison of optimized structures of 2GCzBPPZ and 2GCzBPN at $S_0$ (blue) and $S_1$ (red) states; tert-butyl groups were removed to simplify the calculations. RMSD refers to root-mean-square displacement value.

2GCzBPPZ motivate us to use it as a temperature sensor. 2GCzBPPZ exhibits excellent temperature sensitivity across a very wide temperature range (−70 °C to 70 °C), corresponding to a large color change from yellow at −70 °C through white at room temperature to sky blue at 70 °C. This makes it one of the best TADF-based temperature sensors based on its large dynamic spectral range and associated wide temperature detection range emanating from a single material. Finally, we also employed these two dendrimers as emitters in solution-processed OLEDs to demonstrate their diverse applications.

## Results

### Synthesis
The synthetic routes for 2GCzBPPZ and 2GCzBPN are outlined in Supplementary Fig. 2. Intermediates were synthesized by coupling GCz to the corresponding halogenated acceptor core via a Buchwald-Hartwig C-N cross-coupling or by nucleophilic aromatic substitution reaction in good yields of >80%. Emitters 2GCzBPPZ and 2GCzBPN were obtained by the condensation reaction of intermediates with 1,10-phenanthroline-5,6-diamine (BPPZ) and 1,10-phenanthroline-5,6-dione (BPN), respectively, similar to the synthesis for some of phenazine derivatives[44]. The identity and purity of the two emitters were verified by $^1$H & $^{13}$C NMR spectroscopy, melting point determination, high resolution mass spectrometry and elemental analysis, and high-performance liquid chromatography (HPLC) (Supplementary Figs. 3–18).

### Theoretical calculations
The ground-state ($S_0$) geometries of 2GCzBPPZ and 2GCzBPN were optimized using DFT at the PBE0[45]/6-31 G(d,p) level[46] in the gas phase starting from geometries generated using Chem3D[47]. At the optimized $S_0$ geometries (Supplementary Fig. 19), due to the larger steric hindrance caused by two of the GCz donors substituted on neighboring positions of the BPN acceptor, 2GCzBPN possesses a much more

twisted geometry than that of 2GCzBPPZ, where the torsion angles between donor and acceptor are on average about 67° and 43°, respectively. The HOMOs are localized on the donors while LUMOs are localized on the acceptor group (Fig. 2a and Supplementary Fig. 20). Notably, BPPZ (LUMO = −2.19 eV) is a weaker electron-acceptor than BPN (LUMO = −2.38 eV) (Supplementary Fig. 21). Thus, the HOMO-LUMO gap decreases from 2.87 eV for 2GCzBPPZ to 2.36 eV for 2GCzBPN. The excited-state properties were calculated using time-dependent DFT within the Tamm-Dancoff approximation based on the optimized $S_0$ geometries (Fig. 2b, c)[48,49]. The $S_1$ energies are 2.54 eV for 2GCzBPPZ and 1.95 eV for 2GCzBPN, while the $T_1$ energies are 2.43 to 1.91 eV, respectively, corresponding to $\Delta E_{ST}$ values of 0.11 eV for 2GCzBPPZ and only 0.05 eV for 2GCzBPN, due to its more twisted conformation. Figure 2c displays the natural transition orbitals (NTOs) for $S_1$ and $T_1$. For both compounds, the hole and electron densities are clearly separated for $S_1$, indicating the CT transitions from the ground state. The $T_1$ state of 2GCzBPN is also CT in nature while that of 2GCzBPPZ is better described as one having a mixed LE and CT characters. Reflecting the somewhat stronger overlap between the HOMO and LUMO electron densities, the calculated oscillator strength, $f$, for the $S_0$-$S_1$ transition in 2GCzBPPZ ($f$ = 0.13) is larger than in 2GCzBPN ($f$ = 0.06). To evaluate the geometric rigidity of the compounds, we calculated the root-mean-square displacement (RMSD) value (Fig. 2d) using the VMD program[50] between $S_0$ and $S_1$ geometries each optimized at the PBE0/6-31 G(d,p) level. 2GCzBPN shows a relatively smaller geometry relaxation (RMSD = 0.22 Å) compared to 2GCzBPPZ (RMSD = 0.41 Å), suggesting that 2GCzBPN possesses a more rigid geometry, so nonradiative decay should be relatively attenuated for this compound compared to 2GCzBPPZ.

### Electrochemistry
Next, the energies of the frontier molecular orbitals were inferred from the electrochemical behavior of 2GCzBPPZ and 2GCzBPN by cyclic

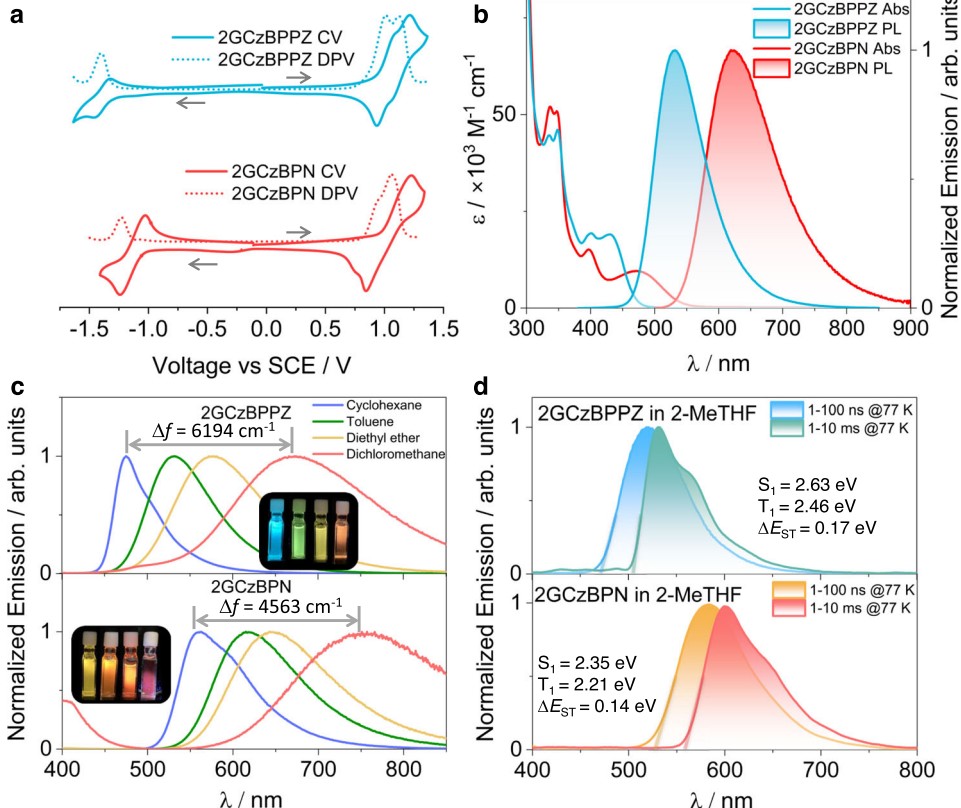

**Fig. 3 | Molecular orbital and excited-state properties in solution. a** Cyclic (CV) and differential pulse voltammograms (DPV) measured in degassed dichloromethane with 0.1 M [$^{n}$Bu$_4$N]PF$_6$ as the supporting electrolyte and Fc/Fc$^+$ as the internal reference (0.46 V vs SCE)[51]. Scan rate = 100 mV s$^{-1}$. **b** UV-vis absorption (Abs) and steady-state photoluminescence (PL) spectra of 2GCzBPPZ and 2GCzBPN recorded in toluene at room temperature ($\lambda_{exc}$ = 340 nm, concentration:

1.6 × 10$^{-5}$ M). **c** PL solvatochromism study of 2GCzBPPZ and 2GCzBPN ($\lambda_{exc}$ = 343 nm). **d** Prompt fluorescence and phosphorescence spectra of 2GCzBPPZ and 2GCzBPN 2-MeTHF at 77 K ($\lambda_{exc}$ = 360 nm, prompt fluorescence and phosphorescence spectra were acquired across a 1-100 ns and 1-10 ms time range, respectively).

voltammetry (CV) and differential pulse voltammetry (DPV) in degassed dichloromethane (DCM) (Fig. 3a). The reduction potentials ($E_{red}$), determined from the DPV peak values, are −1.36 V for 2GCzBPPZ and −1.13 V for 2GCzBPN, respectively (Fc/Fc$^+$ as the internal reference, 0.46 V vs SCE)[51]. The corresponding inferred LUMO energies of −2.98 and −3.21 eV for 2GCzBPPZ and 2GCzBPN, respectively, are consistent with the trend of the energies from the theoretical calculation (Fig. 2a). 2GCzBPPZ and 2GCzBPN both have two resolvable quasi-reversible oxidation waves with $E_{ox}$ of 1.00 and 1.13 V for 2GCzBPPZ, 0.92 and 1.07 V for 2GCzBPN, which correspond to the oxidation of the inner carbazole and the peripheral *tert*-butylcarbazole, respectively[52,53]. Due to the presence of two adjacent donors, 2GCzBPN has a catholically shifted first oxidation potential ($E_{ox}$ = 0.92 V) than that of 2GCzBPPZ ($E_{ox}$ = 1.00 V), corresponding to HOMO levels of −5.26 and −5.34 eV, respectively. The HOMO−LUMO gaps are 2.36 and 2.05 eV for 2GCzBPPZ and 2GCzBPN, respectively, which mirror the trend in the DFT calculated values of 2.87 and 2.36 eV (Fig. 2a).

## Photophysical properties in solution

The UV-Vis absorption spectra of 2GCzBPPZ and 2GCzBPN in dilute toluene are shown in Fig. 3b, and the photophysical properties are summarized in Supplementary Table 1. Both compounds exhibit similar strong absorption profiles centered around 350 nm, which can be attributed to the LE transition of the GCz donors based on a comparison with literature data of GCz[53]. The absorption band at around 400 nm for both compounds is assigned to an LE transition of the acceptor moieties as these align with the absorption of BPN (Supplementary Fig. 22a)[44]. 2GCzBPPZ possesses a stronger ICT absorption at

432 nm ($\varepsilon$ = 19 × 10$^3$ M$^{-1}$ cm$^{-1}$) than that of 2GCzBPN at 475 nm ($\varepsilon$ = 10 × 10$^3$ M$^{-1}$ cm$^{-1}$), due to the slight contribution from the LE character in the state as reflected in Fig. 2c, which is consistent with the calculated higher for 2GCzBPPZ ($f$ = 0.13) than that for 2GCzBPN ($f$ = 0.06). The unstructured and broad photoluminescence (PL) spectra in toluene for both compounds indicate excited states with strong ICT character, with peak maxima, $\lambda_{PL}$, at 534 and 624 nm for 2GCzBPPZ and 2GCzBPN, respectively (Fig. 3b). A strong positive solvatochromism is observed for both compounds (Fig. 3c), which is consistent with the CT nature of the emissive excited state. Furthermore, 2GCzBPPZ exhibits a more significant positive solvatochromism with a $\Delta f$ = 6194 cm$^{-1}$ than that of 2GCzBPN ($\Delta f$ = 4563 cm$^{-1}$), corresponding to the larger transition dipole moment in the excited state, which is corroborated by DFT calculations, for 2GCzBPPZ (3.63 D) than 2GCzBPN (2.83 D). The optical gaps, $E_g$, calculated from the intersection of the normalized absorption and emission spectra (Supplementary Fig. 22b), are 2.62 and 2.26 eV for 2GCzBPPZ and 2GCzBPN, respectively. The prompt fluorescence and phosphorescence spectra in 2-MeTHF glass at 77 K were used to extract the S$_1$ and T$_1$ energies from their respective onsets (Fig. 3d and Supplementary Table 1). The S$_1$ energies of 2GCzBPPZ and 2GCzBPN are 2.63 and 2.35 eV, and the T$_1$ energies are 2.46 and 2.21 eV, respectively. The corresponding $\Delta E_{ST}$ values for 2GCzBPPZ and 2GCzBPN are 0.17 and 0.14 eV, respectively, which align with the calculated results (Fig. 2b). The structured character of the phosphorescence spectra for both emitters implies a T$_1$ state with LE character, corroborated by the calculations indicating mixed $^3$CT/$^3$LE states (LE localized on the acceptor, especially for BPPZ, Fig. 2c). The time-resolved PL (TRPL) decay of 2GCzBPN in degassed

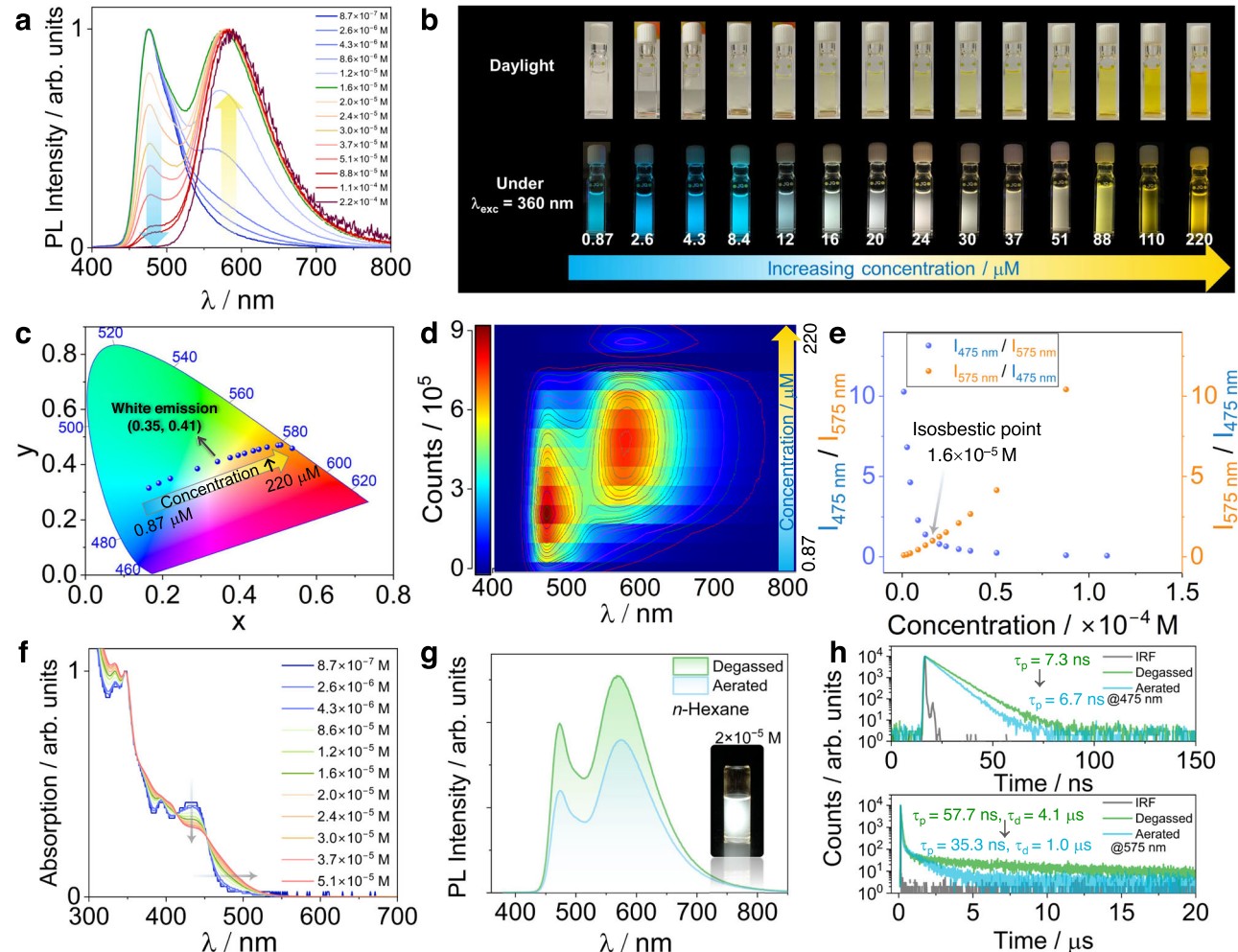

**Fig. 4 | Aggregation-modulated photophysical behavior. a** Concentration-dependent fluorescence spectra for 2GCzBPPZ in *n*-hexane solution ($\lambda_{exc}$ = 340 nm). **b** Corresponding photos under ambient light and using a UV torch ($\lambda_{exc}$ = 360 nm). **c** CIE plot of the color evolution of 2GCzBPPZ as a function of concentration. **d** Concentration-dependent emission mapping. **e** Ratiometric plots of $I_{475}/I_{575}$ (blue dots) and $I_{575}/I_{475}$ (orange dots) vs increasing concentration of 2GCzBPPZ. **f** Concentration-dependent absorption spectra for 2GCzBPPZ in *n*-hexane. **g** PL spectra under degassed and aerated conditions at the concentration of $2.0 \times 10^{-5}$ M in *n*-hexane ($\lambda_{exc}$ = 340 nm). **h** PL decay profiles of the emission $\lambda_{em}$ = 475 nm (top) and $\lambda_{em}$ = 575 nm (bottom) under degassed and aerated *n*-hexane ($\lambda_{exc}$ = 375 nm).

toluene shows biexponential kinetics with a prompt fluorescence lifetime, $\tau_p$, of 24.3 ns and a delayed fluorescence lifetime, $\tau_d$, of 7.5 μs (Supplementary Fig. 23). However, the emission of 2GCzBPPZ decays monoexponentially with $\tau_P$ of 22.3 ns, and no delayed emission is observed. Notably, in both compounds, the emission intensity as well as the prompt lifetime in the toluene solutions were found to be enhanced upon oxygen removal (Supplementary Fig. 23), demonstrating that accessible singlet and triplet states. Similar behavior has been also observed in other reported TADF molecules[54].

### Aggregate modulation

Due to the large degree of π-conjugation inherent in both acceptors, there is likely to be strong intermolecular π–π interactions leading to aggregates. To explore and modulate the emergence of emission from the 2GCzBPPZ and 2GCzBPN aggregates, we measured the PL spectra of 2GCzBPPZ and 2GCzBPN at different concentrations in *n*-hexane solution. As shown in Fig. 4a, there are two emission peaks observed for 2GCzBPPZ where the high-energy emission band at 475 nm converts to the low-energy emission band at 575 nm as a function of increasing concentration of the emitter. We attribute the lower energy emission to its origin from an aggregate based on the different excitation spectra recorded at these two emission peaks (Supplementary

Fig. 24); however, there is negligible change in the emission spectra of 2GCzBPN with increasing concentration, also evidenced by the unchanged Commission Internationale de L´Éclairage (CIE) coordinates (Supplementary Fig. 25). The excitation spectra of 2GCzBPN recorded at different wavelengths further demonstrate the absence of aggregate emission (Supplementary Fig. 26). The corresponding emission color of 2GCzBPPZ evolves from sky-blue to white and then to yellow with increasing the concentration (Fig. 4b), evident in the evolution of the CIE coordinates (Fig. 4c). The relationship between the fluorescence intensity and the concentration in *n*-hexane was analyzed in greater detail (Fig. 4d and Supplementary Fig. 27). First, the integrated emission intensity of the PL spectra increases with increasing concentration due to the combined contribution of emission from monomers and aggregates, and then the intensity decreases as the concentration continues to rise, resulting from aggregation-caused quenching (ACQ). As expected, the emission band at 575 nm gradually intensifies as the concentration increases, owing to the formation of aggregates. The corresponding ratios of the intensity of the emission at 475 to 575 nm ($I_{475}/I_{575}$) and ($I_{575}/I_{475}$) upon increasing concentration are shown in Fig. 4e. An isosbestic point was identified at a concentration of $1.6 \times 10^{-5}$ M where the emission intensity of the monomer ($\lambda_{PL}$ = 475 nm) is equal to that of the aggregates

($\lambda_{PL}$ = 575 nm). Specifically, at low concentrations of 2GCzBPPZ (<$1.6 \times 10^{-5}$ M), the system is dominated by emission from the monomer as molecules are, on average, not sufficiently close to each other to form aggregates to any major extent (Fig. 4e). As the concentration increases (>$1.6 \times 10^{-5}$ M), the probability of intermolecular interactions increases, and the population of aggregates begins to grow. With increasing concentration, the absorption maximum associated with the ICT band of 2GCzBPPZ monomer at ~432 nm gradually decreases along with the emergence of a red-shifted absorption tail from 452 to 550 nm (Fig. 4f), indicating the formation of aggregates through strong intermolecular interactions in the ground state[14,55]. Furthermore, the absorption of 2GCzBPPZ in different solvents at the same concentration also demonstrates that the formation of aggregates occurs in n-hexane but not in higher polarity solvents like toluene, diethyl ether, or DCM (Supplementary Fig. 28). This is likely due to strong solute-solvent interactions that competitively act to suppress the intermolecular interactions between 2GCzBPPZ molecules responsible for aggregate formation[56,57].

Notably, white emission can be obtained in $2 \times 10^{-5}$ M solutions of 2GCzBPPZ in n-hexane as a result of contributions from two distinct emission bands at $\lambda_{PL}$ of 475 and 575 nm (Fig. 4g). TRPL studies in n-hexane under degassed and aerated conditions reveal that the emission band at 475 nm decays with a similar lifetime, $\tau_p$, of around 7.0 ns (Fig. 4h, top), which reflect the quenching of this emission due to fast FRET from the monomers to aggregates, evidenced by the spectral overlap of UV absorption and PL (Supplementary Fig. 29). Under degassed conditions, the emission band at 575 nm decays with biexponential kinetics, with $\tau_p$ of 57.7 ns and $\tau_d$ of 4.1 μs (Fig. 4h, bottom), the delayed emission was largely quenched after exposure to air ($\tau_d$ = 1.0 μs), which could be explained as the quenching of the aggregates-induced TADF in solution[16,58]. The photophysical properties of 2GCzBPPZ as doped films in polymethyl methacrylate (PMMA) were also investigated for comparison. Like the results recorded in n-hexane (Supplementary Fig. 27), the PL efficiency is initially improved as the doping concentration increases, where the highest PL quantum yield ($\Phi_{PL}$) recorded at 1 wt% doping is 72% (Supplementary Table 2), attributed to the combined contribution of the emission from monomers and aggregates. The emission from the TADF aggregates gradually dominates the overall PL spectrum as the concentration increases in the doped films in PMMA (Supplementary Fig. 30a), this associated with a drop in the $\Phi_{PL}$ until it reaches a plateau of approximately 30% as the concentration increases to 50 wt%. However, the $\Phi_{PL}$ of the 2GCzBPPZ neat film is much lower at 11% due to ACQ. This scenario can be alleviated when 2GCzBPPZ is dissolved in toluene at optically dilute concentration, where the $\Phi_{PL}$ is 45% (Supplementary Table 2). The TRPL decays reveal biexponential decay kinetics with associated prompt and delayed components for both emission bands in 0.1 wt% doped film in PMMA. The delayed lifetime $\tau_d$ of the low-energy emission band decreases from 2.12 ms to 150.5 μs with increasing doping concentration from 0.1 wt% to 50 wt% (Supplementary Fig. 30b), while the corresponding prompt lifetime $\tau_p$ slightly increases from 18.7 ns to 25.9 ns with increasing doping concertation. It should be noted that the $\tau_p$ is of the same order of magnitude in n-hexane and PMMA, which indicates that the host has minimal influence on the singlet state radiative and nonradiative decay rates. Therefore, the significant difference in $\tau_d$ by three orders of magnitude in these two media is likely primarily due to variation in the RISC rate. This is determined by the degree of spin-vibronic coupling of the emitter in each host, which is closely related to the relative level of aggregation. The reduced TADF lifetime observed in PMMA at higher doping concentrations can be attributed to the broadened ISC channels resulting from aggregation-induced energy splitting[59–61]. We interpret the absence of a temperature response in PMMA to be due its high glass transition temperature, which restricts the diffusion of 2GCzBPPZ within the matrix.

## Colorimetric temperature sensing

Intrigued by the unusual dual-emissive nature of 2GCzBPPZ, we sought to explore in greater detail the photophysical properties and studied the temperature-dependence of the emission in n-hexane. At room temperature, the $1.6 \times 10^{-5}$ M solution of 2GCzBPPZ is dual-emissive and the sample appears to emit white light (Fig. 5a, b, c), where there are approximately equal contributions from the emission from the monomer (475 nm) and aggregates (575 nm). However, this remarkable temperature-responsive color change was not observed in 2GCzBPN due to the lack of the formation of aggregate emission (Supplementary Figs. 31 and 32). Upon decreasing the temperature towards the solvent freezing point, the low-energy emission band of 2GCzBPPZ increases in intensity dramatically while the high-energy emission band is completely quenched. The corresponding ratio of the intensity of the emission at 575 to 475 nm ($I_{575}/I_{475}$) exponentially decreases with increasing temperature from −70 °C to room temperature ($r^2 = 0.998$) (Fig. 5d). Such an exponential relationship could result from a combination of various factors at low temperatures, like high viscosity and low solubility that will both affect the population of the aggregates[62]. On the other hand, increasing the temperature beyond room temperature reveals a complementary effect where the high-energy emission band becomes more intense and the low-energy emission band all but disappears. In this temperature regime, there is a linear relationship between $I_{475}/I_{575}$ versus temperature ($r^2 = 0.988$ over a temperature region of 25 to 70 °C), corresponding to a ratiometric increase of 6.6% ± 0.2% $K^{-1}$ (Fig. 5e). Overall, 2GCzBPPZ features excellent temperature sensitivity across a broad range of −70 °C to 70 °C, manifested in distinct colorimetric readout from yellow at −70 °C to white at room temperature and finally to sky blue at 70 °C, corresponding to CIE coordinates of (0.50, 0.49) at −70 °C that shift to (0.23, 0.32) associated with blue emission (Fig. 5c and Supplementary Movie 1). Notably, no obvious change in the temperature sensitivity across four cycles was observed (Supplementary Fig. 33). For comparison, we performed experiments using 2GCzBPPZ in toluene and 2-MeTHF (Supplementary Fig. 34). As anticipated, based on its increased solubility in these more polar solvents, a negligible temperature response was observed as aggregation in these solvents was sufficiently inhibited in contrast to the discernible response observed in n-hexane. The temperature-dependent FRET efficiency was then estimated in n-hexane (Supplementary Table 3, Supplementary Fig. 35). The observed weakened FRET efficiency at elevated temperatures implies an increased distance between monomer molecules (Supplementary Fig. 36), and thus fewer aggregates. We thus conclude that the temperature-responsive luminescence results from the variation in the distance between monomers, where at colder temperatures monomers are suitably close that they form aggregates. The broad range of temperature detection coupled with the significant color change exhibited by 2GCzBPPZ makes it a promising temperature sensor, whose properties are much better to previously reported organic fluorescent temperature sensors (Supplementary Table 4)[33,63,64]. Generally, most organic fluorescent temperature sensors rely only on changes in emission intensity with negligible color change and have a narrow temperature detection range, usually between room temperature to ~70 °C[39,40,65]. We have interpreted the origin of the wide dynamic spectral range of our optical temperature sensor to result from a temperature-dependent equilibrium between monomeric and aggregate species.

To provide additional corroboration to the interpretation of the mechanism, molecular dynamics (MD) modeling at different temperatures was executed using the Forcite plus mode in the Materials Studio software (Supplementary Fig. 37)[66,67]. The MD simulations reveal that an increase in the packing distance between adjacent acceptor moieties of 2GCzBPPZ in the aggregate occurs with increasing temperature. In contrast, the MD simulation for 2GCzBPN results demonstrates a negligible temperature-responsive aggregation

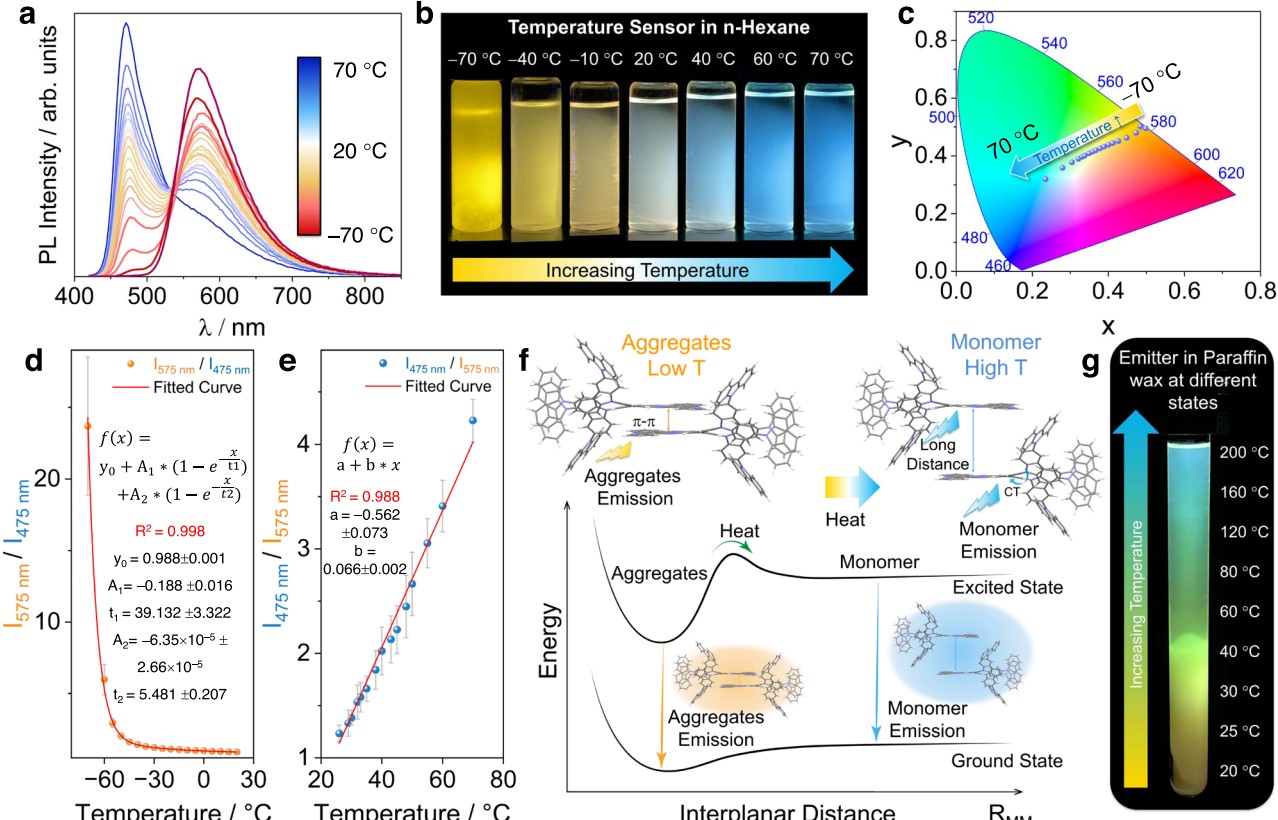

**Fig. 5 | Colorimetric temperature sensing and proposed mechanism.**
**a** Temperature-dependent emission spectra of 2GCzBPPZ in *n*-hexane at a concentration of $1.6 \times 10^{-5}$ M ($\lambda_{exc} = 340$ nm). **b** Photos of 2GCzBPPZ at various temperatures (UV torch $\lambda_{exc} = 360$ nm). **c** CIE plot of the corresponding emission spectra. **d** Ratiometric plot of $I_{575}/I_{475}$ vs temperature upon decreasing the temperature below room temperature. **e** Ratiometric plot of $I_{475}/I_{575}$ vs temperature upon increasing the temperature from room temperature (right). **f** Schematic representation of the thermal response and two-state equilibration model describing the observed abnormal temperature-responsive dual emission phenomenon of 2GCzBPPZ. **g** Spatio-temperature sensor application in paraffin wax (paraffin embedded with 2GCzBPPZ in a test tube (length:160 mm, diameter: 16 mm) excited with a UV torch, $\lambda_{exc} = 360$ nm).

(Supplementary Fig. 38). This is consistent with our hypothesis that temperature-controlled aggregation is the underlying mechanism responsible for the observed PL spectral changes. Thermogravimetric analysis (TGA) and differential scanning calorimetry (DSC) corroborate the interpretation from the MD simulation. 2GCzBPPZ exhibits remarkable thermal stability, with a decomposition temperature reaching nearly 500 °C (Supplementary Fig. 39). The DSC curve shows a broad descending range from 100 to 300 °C, indicative of disruption of short-range ordered molecular packing[68], correlated with color changes in the PL (Fig. 5a). Based on the combined results from the photophysical study, the MD simulations, and the DSC measurements, we can confidently conclude that the temperature sensing mechanism originates from variation in the degree of aggregation (specifically variation in distance between monomers) as a function of the temperature, as schematically shown in Fig. 5f, where upon increasing the temperature, the π-stacking interactions necessary for aggregate formation are disrupted and the monomer population increases, reflected in the emergence of blue emission.

Recognizing the potential of this compound to act as a temperature sensor, we translated its properties into the solid state by embedding the compound into paraffin. As shown in Fig. 5g and Supplementary Movie 2, when photoexcited at 360 nm, the solid paraffin emits in yellow at room temperature. As the temperature increases from 20 to 80 °C, paraffin starts to melt at around 50 °C, and the emission gradually blue-shifts from yellow to green. When the temperature increases beyond 160 °C, the liquid paraffin emits in blue, emulating the emission observed in n-hexane beyond 60 °C. This

distinctive performance in paraffin makes it ideal as a spatio-temperature probe. As demonstrated in Fig. 5g and Supplementary Movie 2, the 2GCzBPPZ-embedded paraffin was melted and poured into a test tube (length:160 mm, diameter: 16 mm) and then allowed to solidify upon cooling. At room temperature, the entire test tube with the solid paraffin shows yellow emission. As the top part of the temperature is selectively heated and the local temperature increases, this part of the paraffin sample exhibits a noticeable color change, while the bottom section of the paraffin maintains its yellow emission as this part is still at room temperature. We have correlated the optical response of the 2GCzBPPZ-embedded paraffin with an external temperature probe to demonstrate that it can act as an accurate temperature sensor (Supplementary Fig. 40, Supplementary Movie 3). In an attempt to use a non-polar solid host with a higher melting temperature, we also incorporated 2GCzBPPZ into 4,4'-bis(9-carbazolyl)-biphenyl (CBP). However, despite having a similar polarity to paraffin, dual emission was not observed in this host. Although the doped film of 2GCzBPPZ in PMMA exhibits concentration-dependent dual emission (Supplementary Fig. 30a), there was no temperature-dependent emission observed, which implies that in this host it is not possible to change significantly intramolecular distances as a function of temperature. The fact that the optical thermometer phenomenon works in paraffin but not the other hosts may be due to the paraffin having a greater thermal expansion coefficient compared to these other host compounds[69]. 2GCzBPPZ thus shows unrivaled temperature sensitivity and with a broad temperature-dependent spectral response compared to previously reported organic temperature sensors[33,39,40,63–65].

**TADF properties in the solid state and exploitation in OLEDs**

To assess the emission properties of 2GCzBPPZ and 2GCzBPN in the solid state, their photophysical properties were investigated in an OLED-relevant host 1,3-bis(*N*-carbazolyl)benzene (mCP) as the host matrix. As 10 wt% doped films in mCP, 2GCzBPN emits at $\lambda_{PL}$ of 601 nm, which is red-shifted compared to 2GCzBPPZ (Supplementary Fig. 41a), which emits at $\lambda_{PL}$ of 531 nm (Supplementary Table. 1). The corresponding $\Phi_{PL}$ values of 2GCzBPN and 2GCzBPPZ are 71 and 57%, respectively, which decrease to 60 and 45%, respectively, under air. The $S_1/T_1$ energies of 2GCzBPPZ and 2GCzBPN inferred from the onsets of the prompt fluorescence and phosphorescence spectra are 2.76/2.50 and 2.38/2.26 eV at 77 K (Supplementary Fig. 41b, c), respectively, thus leading to $\Delta E_{ST}$ values of 0.26 and 0.12 eV, respectively. The room temperature emission from both compounds shows multiexponential decay kinetics, with average prompt fluorescence lifetimes $\tau_p$ of 18.0 and 34.1 ns, and average delayed emission lifetimes, $\tau_d$, of 73.1 and 2.9 μs for 2GCzBPPZ and 2GCzBPN, respectively (Supplementary Fig. 41d). The relative intensities of the delayed PL increased with increasing temperature from 100 K to 298 K, corroborating the TADF nature of the emission of both compounds in the mCP films (Supplementary Fig. 41e, f). To explore the potential of these compounds as emitters in electroluminescent (EL) devices, solution-processed OLEDs were fabricated (see the supporting information). The OLEDs with 10 wt% 2GCzBPPZ and 2GCzBPN in mCP (Supplementary Fig. 42) exhibited green and orange emission with EL maxima, $\lambda_{EL}$, of 552 and 608 nm. The devices with 2GCzBPPZ exhibited a higher maximum external quantum efficiency (EQE$_{max}$) of 15.0% at 581 cd m$^{-2}$ and showed negligible efficiency roll-off at 1000 cd m$^{-2}$ (EQE$_{1000}$ = 14.0%), while the devices with 2GCzBPN showed poorer performance, with EQE$_{max}$ of 5.3%.

## Discussion

Here, we designed two TADF dendrimers 2GCzBPPZ and 2GCzBPN by using different rigid and planar N-doped PAHs as the acceptors combined with two second-generation tercarbazole donor dendrons. Due to the less twisted geometry adopted and the use of the large π-conjugation acceptor, 2GCzBPPZ shows an unusual white emission in solution that results from dual emission from a combination of monomer and aggregates emitting, respectively, at 475 and 575 nm. The dual emission behavior of 2GCzBPPZ is quite sensitive to both the concentration and temperature, which was exploited to demonstrate colorimetric temperature sensing in solution. We also exploited this dual emission behavior in a temperature sensor by embedding 2GCzBPPZ in paraffin. To the best of our knowledge, 2GCzBPPZ shows the broadest spectra and temperature response of any organic temperature sensor. Due to its more twisted geometry, 2GCzBPN shows efficient TADF and there is neither significant aggregate formation nor dual emission. Finally, we also demonstrated the utility of these two compounds as emitters in solution-processed OLEDs, with the device with 2GCzBPPZ achieving an EQE$_{max}$ of 15.0% at $\lambda_{EL}$ of 552 nm.

## Methods

### Synthetic chemistry

Compounds 2GCzBPPZ and 2GCzBPN were synthesized based on Buchwald-Hartwig C-N cross-coupling or nucleophilic aromatic substitution reactions. Detailed procedures are provided in the Supplementary Information.

### Theoretical calculations

Density functional theoretical (DFT) calculation and time-dependent density functional theoretical (TDDFT) calculations were performed using Gaussian 16[70] software in the gas phase. Ground-state geometries were optimized employing the PBE0[45] functional with the Pople 6-31 G(d,p) basis set[46]. Transitions to excited singlet states and

triplet states were calculated using TDDFT within the Tamm-Dancoff approximation (TDA) based on the optimized ground-state geometries[48,49]. Molecular orbitals were visualized with GaussView 6.0[71] and Silico 2.1, an in-house built software package[50,72–78]. Natural transition orbital analysis was conducted using the Multiwfn program[79], and the corresponding molecular orbitals were visualized using VMD program[50].

Molecular dynamic simulation was executed using Material Studio software, incorporating the Forcite plus module[66,67]. Four molecules were constructed and geometrically optimized through molecular dynamic simulations utilizing the Smart algorithm and the COMPASS forcefield[66,67]. Convergence tolerances for energy, force, and displacement were set at 10$^{-4}$ kcal/mol, 0.005 kcal/mol/Å, and 5 × 10$^{-5}$ Å, respectively. A cutoff distance of 15.5 Å for both van der Waals and electrostatic forces was set for calculating noncovalent interactions. Temperature-dependent molecular dynamic simulations were conducted using the COMPASS forcefield and the number-volume-temperature-constant (NVT) ensemble with a time step of 1.5 fs and a total simulation time of 150 ps. Each motional conformation was extracted every 1000 steps, consisting of 101 conformations in the relaxation process. The radial distribution function, g(r), as a function of atomic distance between acceptors in the aggregate, r, was simulated based on the average value of 50 conformations spanning from 75 to 150 ps in the relaxation process, achieving a stable conformational state of molecular aggregates.

### TGA/DSC measurements

TGA and DSC were conducted on a Mettler TGA/DSC3 with a heating rate of 10 K/min under a nitrogen atmosphere. The samples were heated from 50 to 550 °C and the thermal decomposition ($T_d$) was determined at a threshold of 5% weight loss. DSC was performed with a Mettler DSC3 in pierced Al pans, also at 10 K/min under a nitrogen atmosphere.

### Electrochemistry

CV and DPV analyses were performed on an Electrochemical Analyzer potentiostat model 620E from CH Instruments. Samples were prepared in DCM solutions, which were degassed by sparging with DCM-saturated nitrogen gas for 5 min prior to measurements. All measurements were performed using 0.1 M tetra-*n*-butylammonium hexafluorophosphate, [$^n$Bu$_4$N]PF$_6$, in DCM. An Ag/Ag$^+$ electrode served as the reference electrode, a platinum electrode was used as the working electrode and a platinum wire was used as the counter electrode. The redox potentials are reported relative to a saturated calomel electrode (SCE) with a ferrocene/ferrocenium (Fc/Fc$^+$) redox couple as the internal standard (0.46 V vs SCE)[51]. The HOMO and LUMO energies were calculated using the relation $E_{HOMO/LUMO} = -(E_{ox}/E_{red}$ (versus Fc/Fc$^+$) + 4.8) eV[80], where $E_{ox}$ and $E_{red}$ are the anodic and cathodic peak potentials obtained from DPV, respectively, measured versus Fc/Fc$^+$.

### Photophysical measurements

All samples were prepared in HPLC grade *n*-hexane, cyclohexane, toluene, diethyl ether, DCM, 2-MeTHF with varying concentrations in the range of 10$^{-5}$ or 10$^{-6}$ M for absorption and emission study. Absorption spectra were recorded at room temperature using a Shimadzu UV-2600 double beam spectrophotometer. Molar absorptivity determination was verified by linear least-squares fitting of values obtained from at least five independent solutions at varying concentrations with absorbance ranging from 7.5 × 10$^{-6}$ to 3.0 × 10$^{-5}$ mol mL$^{-1}$ for 2GCzBPPZ, 3.6 × 10$^{-6}$ to 3.2 × 10$^{-5}$ mol mL$^{-1}$ for 2GCzBPN.

Degassed solutions were prepared via three freeze-pump-thaw cycles prior to emission analysis using an in-house adapted fluorescence cuvette, purchased from Starna. Steady-state emission and time-resolved emission spectra were recorded at 298 K using an Edinburgh

Instruments FS5 fluorimeter. All samples for the steady-state measurements were excited using a Xenon lamp. Phosphorescence emission spectra were collected with a 5 W microsecond flash lamp. Prompt and delayed fluorescence lifetimes were measured using a picosecond pulsed diode laser (EPL-375).

An integrating sphere (SC-30 module on FS5 fluorimeter) was employed for the $\Phi_{PL}$ measurements of solid samples. The $\Phi_{PL}$ of the films were then measured in air and $N_2$ environment by purging the integrating sphere with $N_2$ gas flow.

### Fitting of the time-resolved luminescence measurements

Time-resolved PL measurements were fitted to exponential decay models, with chi-squared ($\chi 2$) values of between 1 and 2, using the EI FS5 software. Each component of the decay is assigned a weight, ($w_i$), representing the contribution of each component to the total emission.

The average lifetime was then calculated as follows:
For a two exponential decay model:

$$\tau_{AVG} = \tau_1 w_1 + \tau_2 w_2$$

The weights are defined as $w_1 = \frac{A1\tau_1}{A1\tau_1 + A2\tau_2}$ and $w_2 = \frac{A2\tau_2}{A1\tau_1 + A2\tau_2}$, where A1 and A2 are the preexponential-factors for each component.

For a three exponential decay model:

$$\tau_{AVG} = \tau_1 w_1 + \tau_2 w_2 + \tau_3 w_3$$

with weights defined as $w_1 = \frac{A1\tau_1}{A1\tau_1 + A2\tau_2 + A3\tau_3}$, $w_2 = \frac{A2\tau_2}{A1\tau_1 + A2\tau_2 + A3\tau_3}$ and $w_3 = \frac{A3\tau_3}{A1\tau_1 + A2\tau_2 + A3\tau_3}$, where A1, A2 and A3 are the preexponential-factors for each component.

### Temperature sensing measurement

We independently verified the temperature between 20 and 200 °C using a Heidolph 509-67910-00 Pt 1000 temperature sensor, which has an accuracy temperature setting of ±1 °C. The low temperature range (−70 to 20 °C) was monitored using LEYBOLD low-temperature thermometer with an accuracy of ±1 °C.

### OLED characterization

Luminance-current-voltage characteristics were measured in an ambient environment using an IV Curve Measurement System manufactured by Ossila. Electroluminescence spectra were recorded with an Andor DV420-BV CCD spectrometer. OLED emission intensity was detected using a photodiode, with the output current converted to voltage by a transimpedance amplifier. The amplifier's output voltage, designed to be linear to the incident optical power, was measured by a multimeter (Keithley 2000). Using the EL emission spectra, photodiode voltages, and the photodiode's response function, the total photon numbers emitted from the OLEDs were estimated, assuming a Lambertian emission pattern.

## Data availability

The research data supporting this publication have been deposited in the University of St Andrews Research Portal (https://doi.org/10.17630/c4d23206-699a-4378-af84-26c32e028397) and are available from the authors upon request. The optimized molecular coordinates from theoretical calculations are provided as source data.

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

## Acknowledgements

C.S. thanks the China Scholarship Council (201806890001). T.W. acknowledges the support from the European Union's Horizon 2020 research and innovation program under the Marie Skłodowska-Curie grant agreement No. 897098 (AIE-RTP-PLED). D.S. acknowledges support from the Royal Academy of Engineering Enterprise Fellowship (EF2122-13106). We thank the Engineering and Physical Sciences Research Council for support (EP/R035164/1 and EP/W007517/1 to E.Z-C). We thank Dr. Biju Basumatary and Prof. A.P. de Silva for helpful discussions.

## Author contributions

C.S., D.S., and E.Z-C. conceived and designed the project. C.S. synthesized all the related target compounds. C.S. and Y.X. performed the relevant photophysical measurements. C.S., T.W., and D.L. conducted the theoretical calculations. D.S. fabricated the OLEDs. C.S., T.W., and E.Z.C. wrote and edited the manuscript. All authors discussed the results and commented on the manuscript.

## Competing interests

C.S., D.S., and E.Z-C. have filed a patent application GB2315830.6 covering this material and its use as a temperature sensor and emitter for OLEDs, and C.S., D.S., E.Z.-C. declare no other competing interests. T.W., Y.X., and D.L. declare no competing interest.
