## [Peer Review File · Nature Communications]

A Temperature Sensor with a Wide Spectral Range Based on a Dual-emissive TADF Dendrimer SystemREVIEWER COMMENTS

Reviewer #1 (Remarks to the Author):

This manuscript was originally published on Chemrxiv in 2023 and based on novelty it is not clear why it has been sent out for Review by Nature Communications. Zysman-Colman has previously published TADF-based dendrimers with much higher OLED efficiencies (Advanced Materials, 2022, 34, 2110344) than reported here and hence the OLED results add little value to the OLED literature. Indeed, it would almost appear that the dendrimers were originally prepared for application in OLEDs and when not successful, then tried in other applications. While there is nothing wrong with this approach there really does need to be some degree of novelty or substantial insight to be published in a high-end journal. Indeed, the authors spend significant amounts of time describing previous literature on TDAF temperature thermometers, aggregation induced emission (it should be noted that dendrimers are often designed to decrease aggregation of the emissive chromophores) and so on, but the question that is left hanging is why bother making dendrimers if the key features can be achieved using non-dendrimeric materials. There is no disputing that the authors have done every measurement they could think of and some, but the manuscript reads more like a thesis dump than a paper with a defined focus, with much of the critical data shunted off into the Supporting information, which makes it almost unreadable. So, what is the key selling point? Is it simply materials that have a slightly larger temperature range when used as a thermometer? If this is indeed the key point, then much of the data deposited in Supporting information is superfluous.

Further issues:

1. Why were these two molecules chosen? Was the intention to complex them with metal cations through the phenanthroline moiety in a similar manner to that in the literature (Advanced Optical Materials, 2023, 11, 2202720)?
2. It should be noted that electrochemistry does not provide HOMO or LUMO energy levels. Thus, the electrochemistry can provide the electrochemical gap and not the HOMO-LUMO energy gap.
3. Is double hump a scientific description?
4. Why have the authors assumed a 25% outcoupling from their bottom emitting devices?
5. OLED characteristics show that charge injection and/or transport are unbalanced - the EQE increases with increasing drive voltage? The authors should measure the mobilities of the different layers.
6. The ^1H NMR of compound 1 shows it is impure and there is no attempt to assign the protons. It is also missing IR.
7. The ^1H NMR of compound 2 shows it is impure and it is not correctly reported nor are the protons assigned.
8. The ^1H NMR of compound 2GCzBPPZ appears to contain impurities and it is not correctly reported (e.g., coupling constants are not matched) nor are the protons assigned. It is also missing UV-vis data.
9. The ^1H NMR of compound 2GCzBPN appears to be pure but is not correctly reported (e.g., coupling constants are not matched) nor are the protons assigned. It is also missing UV-vis data.

If the publication is to be accepted on the basis of the thermometer responses, then the extraneous data that is not directly related to those results should be removed.

Reviewer #2 (Remarks to the Author):

The researchers introduce a novel temperature sensor molecular design concept by employing a dual-emissive TADF dendrimer system. By adjusting the molecular structure, they found that compound 2GCzBPN, with a highly twisted geometry, displays TADF behavior, whereas 2GCzBPPZ, which has a less twisted structure, exhibits dual emission with a peak at 475 nm attributed to the monomer and another at 575 nm associated with TADF from aggregates. This dual emission is both concentration-dependent and temperature-dependent in solution. 2GCzBPPZ can function as a temperature sensor, showing excellent sensitivity over a wide temperature range in n-hexane. Moreover, solution-processed OLEDs based on these dendrimer emitters demonstrate promising performance, achieving a threefold increase in EQEmax of 15.0% for the device containing 2GCzBPPZ compared to 5.3% for the device with 2GCzBPN. The study is well-executed, and the manuscript is well-structured. Therefore, I strongly recommend that this work be published with minor revisions as follows:

(1) The author stated, “corresponding to a large color change from yellow at -70 °C through white 5 at room temperature to sky blue at 70 °C, which is to the best of our knowledge one of the best TADF-based temperature sensors based on its large dynamic spectral range and associated wide temperature detection range emanating from a single material”. It is recommended to reference and compare relevant literature in support of this claim.

(2) The author provided a comprehensive overview of the strategies used in designing D-A type TADF molecules with dual emissions. Moreover, previous studies have also highlighted the crucial role of the bridging unit between the D and A moieties in influencing dual emissions (refer to: *Angew. Chem. Int. Ed.* 2022, 61, e202116681). It is recommended that the authors delve into discussions regarding this aspect.

(3) The rate of reverse intersystem crossing for 2GCzBPN is 20 times faster than that of 2GCzBPPZ. Interestingly, devices based on 2GCzBPPZ exhibit a substantially higher maximum external quantum efficiency of 15.0% compared to devices utilizing 2GCzBPN (5.3%) as shown in Figure 6d. It is advisable for the authors to provide a more in-depth discussion on this observation.

Reviewer #3 (Remarks to the Author):

The article submitted by Si et al. reports on newly synthesized TADF dendrimers that exhibit diverse temperature-dependent photophysical properties. 2GCzBPPZ is highlighted for its excellent temperature sensing properties. To elucidate the mechanism behind this sensing, the photophysical properties of 2GCzBPPZ have been thoroughly investigated, and the data and arguments presented are compelling. Although I do not see the necessity for the OLED section, the authors simply demonstrate that the reported dendrimers can be used as light emitters in OLEDs. The authors have shown a notable result regarding the durability of the temperature-dependent PL properties in Fig. S31. I believe that this manuscript has the potential to be published in Nature Communications, provided my following requests are addressed.

-The authors mention several times that the enhancement of emission intensity through oxygen removal is attributed to the contribution of the triplet state. However, this alone is not a sufficient condition, as the excited singlet state can also be quenched by oxygen, leading to a decrease in PL intensity (Notsuka, et al., J. Phys. Chem. Lett., 2020, 11, 2, 562–566). The authors should incorporate this reference and modify the discussion accordingly.

-2GCzBPN is characterized as having negligible thermal sensing properties, as shown in Fig. S25. However, the data in Fig. S25 appear to be influenced by temperature. Since the detailed photophysical properties and mechanics of 2GCzBPPZ have been investigated, it would also be beneficial for readers if the same were done for 2GCzBPN. For instance, molecular distance is identified as a critical parameter for thermal sensing, and the authors have used MD simulations for 2GCzBPPZ to support their discussion. Accordingly, MD simulations for 2GCzBPN should demonstrate minimal or no temperature dependence in molecular distance. Additionally, the temperature-dependent photophysical properties of 2GCzBPN should be discussed in terms of electronic states and structure.

-Considering the broad readers in Nature Communications, an example of actual potential application of 2GCzBPPN as thermal sensors should be described.

Response to Reviewers

Reviewer #1 (Remarks to the Author)

General Comment: This manuscript was originally published on Chemrxiv in 2023 and based on novelty it is not clear why it has been sent out for Review by Nature Communications. Zysman-Colman has previously published TADF-based dendrimers with much higher OLED efficiencies (*Advanced Materials*, 2022, 34, 2110344) than reported here and hence the OLED results add little value to the OLED literature. Indeed, it would almost appear that the dendrimers were originally prepared for application in OLEDs and when not successful, then tried in other applications. While there is nothing wrong with this approach there really does need to be some degree of novelty or substantial insight to be published in a high-end journal.

Our response: As the reviewer points out, our previous work has demonstrated the potential of TADF dendrimers as emitters in high-efficiency green solution-processed OLEDs, SP-OLEDs (*Adv. Mater.* **2022**, 34, 2110344). In that work, the di-*tert*-butylcarbazole donor dendrons substituted meta to the triazine acceptor were introduced to suppress aggregation-caused quenching and enhance RISC. Here, we demonstrate that the same donor dendrons can decorate a much stronger acceptor to access yellow and red emitters suitable for SP-OLEDs at these colour points. Despite the lower efficiency SP-OLEDs, we presented the results to demonstrate the broader utility of this type of TADF dendrimer design for use SP-OLEDs.

While undertaking our photophysical study in advanced of SP-OLED fabrication, we observed an unusual behavior in apolar solvents that we believe is very impactful. This behaviour relates to the changes in emission colour as a functional of temperature, showing an outstanding and very broad temperature sensing range for an organic optical temperature sensor. The temperature sensing results from the first observation of the control of monomer to aggregate emission from TADF dendrimers in solution (and paraffin). The dual emission mechanism differs significantly from prior reports such as those in *Adv. Mater.* **2020**, 32, 1903269; *J. Am. Chem. Soc.* **2006**, 128, 14081-14092; *Adv. Sci.* **2020**, 7, 2001845; *J. Phys. Chem. Lett.* **2021**, 12, 1162–1168; *Angew. Chem. Int. Ed.* **2023**, 62, e2023021). It should be noted that this is the first report of *single-component* TADF emitters for ratiometric temperature sensing, discernible by the naked eye.

Of the reported organic emitter temperature sensors, most are based on host:dopant systems (see Table S4), which do not present a homogenous luminescence as a result of the heterogeneity of the guest doping in the film and possible phase separation. In our recent work, we have demonstrated that dual phosphorescence can serve as low-temperature sensing based on afterglow evolution from

two different triplet excited states in a single emitter, with a sensing range spanning from -196 to 25 °C (*Angew. Chem. Int. Ed.* **2022**, *61*, e202206681; *Angew. Chem. Int. Ed.* **2023**, *62*, e202309718). These temperature sensors, however, fail to function at higher temperatures. This dilemma has been overcome by employing our TADF dendrimer optical temperature in this work, which shows an exceptionally broad temperature sensing range (-70–200 °C). Thus, we believe that all of these points both demonstrate the novelty of our temperature sensor and its impact; of course, there is also novelty in terms of the molecular design.

Indeed, the authors spend significant amounts of time describing previous literature on TDAF temperature thermometers, aggregation induced emission (it should be noted that dendrimers are often designed to decrease aggregation of the emissive chromophores) and so on, but the question that is left hanging is why bother making dendrimers if the key features can be achieved using non-dendrimeric materials.

Our response: We agree with the reviewer that dendrimers generally exhibit anti-aggregation behaviour due to the protection of peripheral dendrons. Indeed, we do not observe any aggregation in more polar media. What is particularly interesting and impactful is how this aggregation behaviour exists only in certain apolar media. This occurs because the solubility profile of the emitters is suitably poor in these media that they begin to aggregate, forming π -stacking interactions between the acceptor moieties of adjacent emitters (Figures **5f**, **S36** and **S37**). We exploit this weak interaction in our optical sensors by demonstrating that this interaction can be broken at more elevated temperatures, thus restoring the monomer emission, which is blue-shifted compared to the aggregate.

We endeavour to demonstrate the novelty and impact of our optical temperature probe by providing a rather comprehensive overview in the introduction of the prior art of TADF thermometers and AIE, which permits the reader to appreciate what is distinct about our present study. Most TADF thermometers using non-dendrimeric materials that are highly dependent on emission lifetime, intensity, and color variation. These non-ratiometric temperature sensors inherently face challenges in terms of their ability to accurately detect temperature fluctuations. So far, there are relatively few examples of dual-emissive TADF systems, particularly those exhibiting both monomer and aggregate emission in the solution state. In this work, we have achieved, for the first time, monomer and aggregate dual emissions based on TADF dendrimers, along with a very broad temperature sensing range.

There is no disputing that the authors have done every measurement they could think of and some, but the manuscript reads more like a thesis dump than a paper with a defined focus, with much of the critical data shunted off into the Supporting information, which makes it almost unreadable. So, what is the key selling point? Is it simply materials that have a slightly larger temperature range when used as a thermometer? If this is indeed the key point, then much of the data deposited in Supporting information is superfluous.

Our response: We thank the reviewer for recognizing the effort involved in this work. In our revision, we have attempted to make clearer the key selling points of the paper, which is that this is the first report of a *single-component* TADF dendrimer used for ratiometric temperature sensing,

and that this material shows an especially broad temperature sensing range and highly sensitive colour response. For clarity, we have summarized the novelty from the following perspectives:

- 1) **Novelty of the emitter design:** These two compounds are new. Although they share the same donor dendrons and have similar though distinct acceptors, they exhibit completely different photophysical properties due to the different conformations adopted. This is a significant insight highlighted in our paper.
- 2) **Applications:** TADF materials have garnered significant attention in the field of organic electronics, particularly as emitters in OLEDs. This has motivated us to include OLED data within this manuscript for completeness. However, the application with the most potential impact is their use as optical temperature sensors, which is why we focus considerable space to demonstrating their utility and the underpinning photophysics.
- 3) **Performance:** We have quantitatively demonstrated a sensing range from -70 to 70 °C in n-hexane and from 20 to 200 °C in paraffin wax. This represents a remarkable temperature response of 6.6% K⁻¹, which is significant compared to existing fluorescent temperature sensors, as shown in Table S4.
- 4) **Mechanism:** The mechanism by which the temperature sensing occurs differs from other materials in prior reports and relies on the control of the speciation between emissive monomers and aggregates, which can only occur in certain apolar media.

To ensure clarity and better reflect the focus of our current study, as reflected by the manuscript title, we have simplified the discussion in the OLED section and provided detailed information in the supporting information.

Further issues

Comment 1: Why were these two molecules chosen? Was the intention to complex them with metal cations through the phenanthroline moiety in a similar manner to that in the literature (*Advanced Optical Materials*, 2023, 11, 2202720)?

Our response: Regarding the molecular design, the donor dendrons were employed to delocalize the electron density in the HOMO, thereby reducing ΔE_{ST} (*Nat. Mater.* **2015**, *14*, 330-336) and enable efficient TADF. The planar phenazine-based acceptor was anticipated to (1) promote a red-shifted emission compared to our prior work in SP-OLED emitter design and (2) confer temperature-responsive molecular aggregation via π - π interactions to the TADF dendrimers. In this work, we have demonstrated the validity of this design through theoretical simulations, structural characterization, and photophysical investigations.

As the reviewer pointed out, the phenanthroline moiety possesses the capability to coordinate with metals. Indeed, in our recent work, we have demonstrated the utility of phenanthroline-derived emitters in sensing metal ions (*Adv. Funct. Mater.* **2024**, 2315935).

Comment 2: It should be noted that electrochemistry does not provide HOMO or LUMO energy levels. Thus, the electrochemistry can provide the electrochemical gap and not the HOMO-LUMO energy gap.

Our response: We agree with the reviewer that the HOMO and LUMO energy levels cannot technically be measured directly via electrochemistry. However, it is widely recognized and reported that redox potentials determined in solution by electrochemical measurements can be used

to estimate HOMO/LUMO levels by considering the vacuum energy level of ferrocene to be 4.8 eV (other values have also been used, so what matters is that the methodology for conversion of redox data to HOMO/LUMO levels is explicitly provided), the HOMO and LUMO energies can be inferred using the relation: $E_{\text{HOMO/LUMO}} = -(E_{\text{ox}}/E_{\text{red}} \text{ (versus Fc/Fc}^+) + 4.8) \text{ eV}$ (*Adv. Mater.* **1995**, *7*, 551-554). The detailed description related to the electrochemistry measurements and the conversion methodology have already been included in the supporting information.

We have updated the text, and it now reads, “The reduction potentials (E_{red}), determined from the DPV peak values, are -1.36 V for **2GCzBPPZ** and -1.13 V for **2GCzBPN**, respectively (Fc/Fc⁺ as the internal reference, 0.46 V vs SCE).⁵¹ The corresponding inferred LUMO energies of -2.98 eV and -3.21 eV for **2GCzBPPZ** and **2GCzBPN** ($E_{\text{LUMO}} = -(E_{\text{red}} \text{ (versus Fc/Fc}^+) + 4.8) \text{ eV}$), respectively, are consistent with the trend of the energies from the theoretical calculation (Figure 2a).”

Comment 3: Is double hump a scientific description?

Our response: We have updated the text, and it now reads, “Both compounds exhibit similar strong absorption profiles centered at around 350 nm , which can be attributed to locally excited (LE) transitions of the GCz donors based on a comparison with literature data of GCz.⁵³”

Comment 4: Why have the authors assumed a 25% outcoupling from their bottom emitting devices?

Our response: The light outcoupling efficiency, η_{out} , is defined as the ratio of the number of photons that exit the OLED to the total number of photons generated within the emitting layer (EML) (*Adv. Funct. Mater.* **2005**, *15*, 1839–1844). This is influenced by several factors, including the thicknesses and refractive indices of the functional material layers as well as the direction of the photon generated during exciton relaxation (*Adv. Funct. Mater.*, **2019**, *29*, 1808803.). In an OLED, the light generated by the radiative recombination of excitons can be trapped or lost due to several factors: surface plasmons (~40%), waveguide (~10%) and substrate modes (~30%) as shown in Figure S41. Notably, the waveguided and substrate modes are mainly influenced by the reflective indices of materials, thereby leading to η_{out} of 20~30% (*Adv. Mater.* **2021**, *33*, 2100677; *J. Photonics Energy* **2015**, *5*, 057607). Hence, a common conservative assumption is 25%. We have updated the text in ESI, and it reads “Considering the measured Φ_{PL} (Table S1) and assuming 25% outcoupling efficiency associated with an isotropic orientation of the transition dipole moment of the emitter,^{45,46} the EQE_{max} for devices with **2GCzBPPZ** and **2GCzBPN** were expected to be 14.3% and 17.8%, respectively.”

Figure S42. Schematic illustration of the light extraction from OLEDs.

Comment 5: OLED characteristics show that charge injection and/or transport are unbalanced - the EQE increases with increasing drive voltage? The authors should measure the mobilities of the different layers.

Our response: We agree with the reviewer that charge injection and/or transport in our devices is unbalanced. Achieving charge-carrier balance is indeed one of the main challenges in the development of SP-OLEDs. Based on the reviewers' comments, we have opted to de-emphasize the SP-OLEDs in this paper, moving the majority of the text to the ESI, and to focus on the use of these emitters as optical temperature probes. In this context, though we agree that the devices are not optimized in terms of charge balance, we do not see the value of dedicating more effort to a facet of the study that the reviewers have deemed not so impactful and thus we have not measured mobilities. Rather, future studies will focus on reporting much-improved SP-OLEDs at these colour points.

Comment 6: The ^1H NMR of compound **1** shows it is impure and there is no attempt to assign the protons. It is also missing IR.

Our response: We have purified compound **1** and now report updated ^1H NMR spectra (Figure S2a). We have also recorded the ^1H - ^1H COSY NMR spectrum to confirm the proton assignment (Figure S2b). Combined with HRMS, we believe that we have provided the necessary characterization to confirm both identity and purity. As a matter of course, we do not provide IR data and do not believe that IR analysis provides any significant additional characterization insight for these materials and as such we have opted not to include IR spectra.

“3,6-bis(3,3'',6,6''-tetra-*tert*-butyl-9'*H*-[9,3':6',9''-tercarbazol]-9'-yl)phenanthrene-9,10-dione (1**): Yield: 80%. $R_f = 0.2$ (25% DCM/hexane). $M_p = 341\text{-}343$ °C. ^1H NMR (500 MHz, CDCl_3) δ 8.68 (d, $J = 8.3$ Hz, 2H), 8.46 (s, 2H), 8.26 (s, 4H), 8.13 (s, 8H), 7.99 (d, $J = 9.3$ Hz, 2H), 7.76 (d, $J = 8.7$ Hz, 4H), 7.65 (d, $J = 8.7$ Hz, 4H), 7.41 (d, $J = 8.7$ Hz, 8H), 7.29 (d, $J = 8.6$ Hz, 8H), 1.43 (s, 72H).”**

Figure S2. (a) 1D ^1H NMR and (b) ^1H - ^1H COSY NMR spectra of **1 in CDCl_3 .**

Comment 7: The ^1H NMR of compound **2** shows it is impure and it is not correctly reported nor are the protons assigned.

Our response: We have purified compound 2 and updated the NMR spectra alongside the proton assignment (Figures S5 and S6). Notably, influenced by the double substitution of two dendrons at adjacent sites, separating No. 6 and 7 protons in the ^1H NMR spectrum is challenging. This phenomenon has been reported in our previous work. (*Adv. Mater.* **2022**, 2110344)

5,6-bis(3,3'',6,6''-tetra-*tert*-butyl-9'*H*-[9,3':6',9''-tercarbazol]-9'-yl)benzo[*c*][1,2,5]thiadiazole (2): Yield: 83%. $R_f = 0.4$ (20% DCM/Hexane). **Mp** > 400 °C. ^1H NMR (500 MHz, THF-*d*₈) δ 9.04 (s, 2H), 8.17 (s, 4H), 8.11 (s, 8H), 7.53 (d, $J = 8.6$ Hz, 4H), 7.29 (d, $J = 8.6$ Hz, 4H), 7.08 (s, 16H), 1.32 (s, 72H). ^{13}C NMR (126 MHz, THF-*d*₈): δ 152.77, 140.07, 137.98, 137.76, 129.55, 122.86, 122.57, 121.79, 121.50, 121.23, 116.99, 113.96, 109.67, 106.89, 32.42, 29.58. **HR-MS [M+H]⁺ Calculated:** (C₁₁₀H₁₁₀N₈S) 1574.8574; **Found:** 1574.8568.

Figure S5. (a) 1D ^1H NMR and (b) ^1H - ^1H COSY NMR spectra of **2** in THF- d_8 .

Figure S6. ^{13}C NMR spectrum of **2** in THF- d_8 .

Comment 8: The ^1H NMR of compound 2GCzBPPZ appears to contain impurities and it is not correctly reported (e.g., coupling constants are not matched) nor are the protons assigned. It is also missing UV-vis data.

Our response: We have recorded the HPLC trace of **2GCzBPPZ**, demonstrating its purity to be greater than 98.6%. This is of sufficient purity to permit relevant photophysical investigations of **2GCzBPPZ**. The updated ^1H NMR spectrum is shown in Figure S8. The coupling constant and the proton assignment have also been updated. The UV-vis absorption data were presented in Figure 3b and discussed in the section of *Photophysical properties in solution*.

“12,15-bis(3,3'',6,6''-tetra-*tert*-butyl-9'*H*-[9,3':6',9''-tercarbazol]-9'-yl)dibenzo[*a,c*]dipyrido[3,2-*h*:2',3'-*j*]phenazine (2GCzBPPZ): Yield: 68%. R_f = 0.3 (20% DCM/hexane). Mp >400 °C. ^1H NMR (500 MHz, CDCl_3) δ 10.05 (d, J = 8.5 Hz, 2H), 9.97 (d, J = 8.2 Hz, 2H), 9.42 (d, J = 4.5 Hz, 2H), 9.07 (s, 2H), 8.32 (d, J = 1.8 Hz, 2H), 8.31 (s, 4H), 8.15 (s, 8H), 7.98 (dd, J = 8.1, 4.4 Hz, 2H), 7.82 (d, J = 8.7 Hz, 4H), 7.66 (d, J = 8.7 Hz, 4H), 7.43 (d, J = 8.8 Hz, 8H), 7.34 (d, J = 8.6 Hz, 8H), 1.44 (s, 72H). ^{13}C NMR (126 MHz, CDCl_3): δ 152.52, 142.72, 140.96, 140.39, 140.14, 139.68, 133.93, 133.16, 131.66, 129.88, 128.79, 127.70, 127.54, 126.40, 124.48, 123.62, 123.22, 121.73, 119.71, 116.26, 111.02, 109.04, 77.26, 77.01, 76.76, 34.72, 32.03. HR-MS $[\text{M}+\text{H}]^+$ Calculated: ($\text{C}_{130}\text{H}_{120}\text{N}_{10}$) 1821.9802; Found: 1821.9893. Anal. Calcd. for $\text{C}_{130}\text{H}_{120}\text{N}_{10}$: C, 85.68%; H, 6.64%; N, 7.69%. Found: C, 85.77%; H, 6.84%; N, 7.68%. HPLC analysis: 98.6%

pure on HPLC analysis, retention time 10.190 minutes in mixture of 85% Acetonitrile and 15% Water.

Figure S8. (a) 1D ^1H NMR and (b) ^1H - ^1H COSY NMR spectra of **2GCzBPPZ** in CDCl_3 .

Comment 9: The ^1H NMR of compound **2GCzBPN** appears to be pure but is not correctly reported (e.g., coupling constants are not matched) nor are the protons assigned. It is also missing UV-vis data.

Our response: We have updated the related NMR spectra and corrected the mistakes in the assignment. Regarding the UV-vis absorption data, as we replied to Comment 8, the original manuscript had already included a discussion.

“11,12-bis(3,3'',6,6''-tetra-*tert*-butyl-9'*H*-[9,3':6',9''-tercarbazol]-9'-yl)dipyrido [3,2-*a*:2',3'-*c*]phenazine (2GCzBPN): Yield: 76%. R_f = 0.3 (25% DCM/Hexane). Mp > 400 °C. ^1H NMR (500 MHz, CDCl_3) δ 9.87 (d, J = 8.1 Hz, 2H), 9.46 (d, J = 3.6 Hz, 2H), 9.23 (s, 2H), 8.11 (s, 8H), 8.05 (s, 4H), 7.98 (dd, J = 8.1, 4.4 Hz, 2H), 7.49 (d, J = 8.6 Hz, 4H), 7.33 (d, J = 8.7 Hz, 4H), 7.09 (s, 16H), 1.37 (s, 72H). ^{13}C NMR (126 MHz, CDCl_3): δ 153.43, 148.91, 142.90, 142.62, 142.22, 139.77, 139.29, 136.28, 134.24, 131.85, 131.53, 127.24, 125.34, 124.61, 124.52, 123.74, 123.13, 118.97, 116.19, 111.17, 108.73, 77.36, 77.11, 76.86, 34.68, 32.05. HR-MS [$\text{M}+\text{H}$] $^+$ Calculated: ($\text{C}_{122}\text{H}_{116}\text{N}_{10}$) 1721.9418; Found: 1721.9457.”

Figure S13. (a) 1D ¹H NMR and (b) ¹H-¹H COSY NMR spectra of 2GCzBPN in CDCl₃.

Comment 10: If the publication is to be accepted on the basis of the thermometer responses, then the extraneous data that is not directly related to those results should be removed.

Our response: We appreciate the reviewer's opinion. As a matter of principle, we do not believe in removing data; however, to better reflect the focus of the manuscript on the optical temperature probes, we have simplified the discussion in the OLED section and moved the majority of the text to the ESI to improve the narrative and clarity of the manuscript.

Reviewer #2 (Remarks to the Author)

General Comment: The researchers introduce a novel temperature sensor molecular design concept by employing a dual-emissive TADF dendrimer system. By adjusting the molecular structure, they found that compound 2GCzBPN, with a highly twisted geometry, displays TADF behavior, whereas 2GCzBPPZ, which has a less twisted structure, exhibits dual emission with a peak at 475 nm attributed to the monomer and another at 575 nm associated with TADF from aggregates. This dual emission is both concentration-dependent and temperature-dependent in solution. 2GCzBPPZ can function as a temperature sensor, showing excellent sensitivity over a wide temperature range in n-hexane. Moreover, solution-processed OLEDs based on these dendrimer emitters demonstrate promising performance, achieving a threefold increase in EQEmax of 15.0% for the device containing 2GCzBPPZ compared to 5.3% for the device with 2GCzBPN. The study is well-executed, and the manuscript is well-structured. Therefore, I strongly recommend that this work be published with minor revisions as follows:

Our response: We thank the reviewer for the positive assessment of our work.

Comment 1: The author stated, "corresponding to a large color change from yellow at -70 °C through white at room temperature to sky blue at 70 °C, which is to the best of our knowledge one of the best TADF-based temperature sensors based on its large dynamic spectral range and associated wide temperature detection range emanating from a single material". It is recommended to reference and compare relevant literature in support of this claim.

Our response: We appreciate the reviewer's suggestion. The text has been updated with cited references and now it reads, "The broad range of temperature detection coupled with the significant color change exhibited by 2GCzBPPZ make it a promising temperature sensor, whose properties are much superior to previously reported organic fluorescent temperature sensors (Table S4).^{33,63,64} Generally, most organic fluorescent temperature sensors rely only on changes in emission intensity with negligible color change and have a narrow temperature detection range, usually between room temperature to ~70 °C.^{39,40,65}"

To support the claim, we have summarized the temperature sensing performance of the previous reports and our work in Table S4 for clarity.

Comment 2: The author provided a comprehensive overview of the strategies used in designing D-A type TADF molecules with dual emissions. Moreover, previous studies have also highlighted the crucial role of the bridging unit between the D and A moieties in influencing dual emissions (refer to: Angew. Chem. Int. Ed. 2022, 61, e202116681). It is recommended that the authors delve into discussions regarding this aspect.

Our response: We thank the reviewer for providing the related reference. We have updated the text and it now reads, “Dual emission could also arise from equilibrated LE and CT excited states.¹¹ Geng *et al.* designed two molecules, **TMCz- σ -TRZ** and **DMAC- σ -TRZ**, that contain a hexafluoroisopropylidene σ -bridging unit between the D and A moieties. Both of these D- σ -A compounds showed simultaneous emission from LE and CT states, the latter of which showed TADF character.¹² Ma *et al.* observed dual emission from **MeCz-BP**, where the short wavelength emission bands originated from emission from a LE state from the MeCz donor and the long wavelength emission band was ascribed to CT emission.¹³”

The literature review study has been updated in Figure S1.

(a) Dual-emission from dual conformations

PTZ-TRZ

a-DMAC-TRZ

J. Phys. Chem. C **2014**, *118*, 15985 *Angew. Chem. Int. Ed.* **2019**, *58*, 582

(c) Dual-emission from hybrid intramolecular and intermolecular CT

CPzP

CPzPO

SPzP

SPzPO

Adv. Funct. Mater. **2017**, *27*, 1703918

(b) Dual-emission from equilibrated LE and CT

DMAC- σ -TRZ

TMCz- σ -TRZ

Angew. Chem. Int. Ed. **2017**, *56*, 16536

MeCz-BP

Angew. Chem. Int. Ed. **2022**, *61*, e202116681

(d) Dual-emission from two ICTs based on asymmetric triad structures

M-1

Angew. Chem. Int. Ed. **2020**, *59*, 17018

Figure S1. Examples of dual emission from (a) dual conformations, (b) equilibrated LE and CT states, (c) “hybrid intramolecular and intermolecular CT” states and (d) two ICT states based on asymmetric triad structures.

Comment 3: The rate of reverse intersystem crossing for 2GCzBPN is 20 times faster than that of 2GCzBPPZ. Interestingly, devices based on 2GCzBPPZ exhibit a substantially higher maximum external quantum efficiency of 15.0% compared to devices utilizing 2GCzBPN (5.3%) as shown in Figure 6d. It is advisable for the authors to provide a more in-depth discussion on this observation.

Our response: The difference in the k_{RISC} of **2GCzBPN** and **2GCzBPPZ** primarily arises from their divergent ΔE_{ST} values. However, the EQE_{max} values of the OLEDs are influenced by multiple factors. The EQE_{max} can be calculated based on the following equation:

$$\text{EQE} = \gamma \cdot \eta_r \cdot \Phi_{\text{PL}} \cdot \eta_{\text{out}}$$

Where γ , η_r , Φ_{PL} , and η_{out} represent the charge carrier injection efficiency, the exciton harvesting efficiency, the photoluminescence quantum efficiency, and the light outcoupling efficiency, respectively. The ideally balanced charge carrier injection should result in $\gamma = 1$. Unbalanced charge carrier injection in SP-OLEDs often makes γ less than 1, dependant heavily on the device configuration. Also, the light outcoupling efficiency is another factor that affects the device performance, which may explain the inferior performance of the device with **2GCzBPN**. However, at this stage it is unclear why the EQE_{max} for **2GCzBPN** is so low. Considering that the OLED investigation is not the focus of this work, we have move detailed discussion of the OLED section to the supporting information, as requested by Reviewer 1, and subsequent studies will focus on improving the performance of SP-OLEDs.

We have updated the text in the ESI and now it reads, “The corresponding rate constants of intersystem crossing (k_{ISC}) for both compounds in mCP films are $1.61 \times 10^7 \text{ s}^{-1}$ and $0.45 \times 10^7 \text{ s}^{-1}$ for **2GCzBPPZ**, and **2GCzBPN**, respectively, while the rate constants of RISC (k_{RISC}) for **2GCzBPN** reached $4.1 \times 10^5 \text{ s}^{-1}$, a value 20 times faster than in **2GCzBPPZ** of $1.93 \times 10^4 \text{ s}^{-1}$, due to the much smaller ΔE_{ST} of **2GCzBPN** (0.12 eV) than that of **2GCzBPPZ** (0.26 eV).⁴⁴”

In the ESI it reads: “This indicates that the device with **2GCzBPPZ** has effectively unity exciton utilization efficiency however, at this stage it is unclear why the EQE_{max} for **2GCzBPN** is so low. We speculate surface plasmons and waveguide and substrate modes are responsible for the light loss (Figure S42). Besides, compared to **2GCzBPPZ**, **2GCzBPN** with its lower linearity and planarity probably results in a lower light outcoupling efficiency.^{47,48}”

Reviewer #3 (Remarks to the Author)

The article submitted by Si et al. reports on newly synthesized TADF dendrimers that exhibit diverse temperature-dependent photophysical properties. **2GCzBPPZ** is highlighted for its excellent temperature sensing properties. To elucidate the mechanism behind this sensing, the photophysical properties of **2GCzBPPZ** have been thoroughly investigated, and the data and arguments presented are compelling. Although I do not see the necessity for the OLED section, the authors simply demonstrate that the reported dendrimers can be used as light emitters in OLEDs. The authors have shown a notable result regarding the durability of the temperature-dependent PL properties in Fig. S31. I believe that this manuscript has the potential to be published in Nature Communications, provided my following requests are addressed.

Our response: We thank the reviewer for the positive assessment of our work. To enhance clarity, we have moved the detailed discussion of the OLED section to the supporting information.

Comment 1: The authors mention several times that the enhancement of emission intensity through oxygen removal is attributed to the contribution of the triplet state. However, this alone is not a sufficient condition, as the excited singlet state can also be quenched by oxygen, leading to a

decrease in PL intensity (Notsuka, et al., J. Phys. Chem. Lett., 2020, 11, 2, 562–566). The authors should incorporate this reference and modify the discussion accordingly.

Our response: We thank the reviewer for providing this useful reference and indeed recognize that O₂ can quench the luminescence of compounds via PET from the singlet excited state of the emitter; however, we do note that significant quenching of the luminescence by O₂ is commonly ascribed to PEnT from the triplet state of the emitter.

“Notably, in both compounds, the emission intensity as well as the prompt lifetime in toluene solutions were found to be enhanced upon oxygen removal (Figure S22), demonstrating that oxygen quenches both accessible singlet and triplet states. Similar behavior has been also observed in other reported TADF molecules.⁵⁴”

Comment 2: 2GCzBPN is characterized as having negligible thermal sensing properties, as shown in Fig. S25. However, the data in Fig. S25 appear to be influenced by temperature. Since the detailed photophysical properties and mechanics of 2GCzBPPZ have been investigated, it would also be beneficial for readers if the same were done for 2GCzBPN. For instance, molecular distance is identified as a critical parameter for thermal sensing, and the authors have used MD simulations for 2GCzBPPZ to support their discussion. Accordingly, MD simulations for 2GCzBPN should demonstrate minimal or no temperature dependence in molecular distance. Additionally, the temperature-dependent photophysical properties of 2GCzBPN should be discussed in terms of electronic states and structure.

Our response: We appreciate the valuable suggestion from the reviewer. We wish to clarify that Figure S26 presents the normalized PL intensity of 2GCzBPN *versus* concentration in *n*-hexane. Unlike the color change behavior of 2GCzBPPZ, 2GCzBPN exhibits a negligible color evolution as its concentration in *n*-hexane rises (Figure S24b). The unchanged excitation spectra of 2GCzBPN as a function of wavelength further reveal the absence of aggregate emission (Figure S25). Additionally, we investigated the temperature-dependent PL behaviour of 2GCzBPN in *n*-hexane (Figure S30). As anticipated, 2GCzBPN does not demonstrate remarkable temperature sensing performance like 2GCzBPPZ due to the lack of the formation of emissive aggregates. We have updated the text, and it reads, “however, there is negligible change in the emission spectra of 2GCzBPN with increasing concentration, also evidenced by the unchanged Commission Internationale de L’Éclairage (CIE) coordinates (Figure S24). The excitation spectra of 2GCzBPN recorded at different wavelengths further demonstrate the absence of aggregate emission (Figure S25)” and “At room temperature, the 1.6×10⁻⁵ M solution of 2GCzBPPZ is dual-emissive and the sample appears to emit white light (Figures 5a and 5b), where there are approximately equal contributions from the emission from the monomer (475 nm) and aggregates (575 nm). However, this remarkable temperature-responsive color change was not observed in 2GCzBPN due to an absence of the formation of aggregate emission (Figures S30 and S31).”

Figure S24. (a) Concentration-dependent emission spectra for **2GCzBPN** in *n*-hexane solution ($\lambda_{\text{exc}} = 340$ nm); (b) CIE plot of the color evolution of **2GCzBPN** as a function of concentration.

Figure S25. Excitation spectra of **2GCzBPN** recorded at $\lambda_{\text{PL}} 560$ and 592 nm in *n*-hexane at 298 K.

Figure S30. (a) Temperature-dependent emission spectra and (b) normalized PL of **2GCzBPN** in *n*-hexane at a concentration of 1×10^{-5} M ($\lambda_{\text{exc}} = 340$ nm).

Figure S31. CIE plot of the temperature-dependent emission spectra of **2GCzBPN** in *n*-hexane at a concentration of 1×10^{-5} M ($\lambda_{\text{exc}} = 340$ nm).

To elucidate the temperature sensing mechanism, we further conducted MD simulations for **2GCzBPN**. The simulation results demonstrate a negligible temperature-responsive aggregation, where the radial distribution function $g(r)$ does not change significantly with temperature. The updated text reads, “The MD simulations reveal that there is an increase in the packing distance between adjacent acceptor moieties of **2GCzBPPZ** in the aggregate with increasing temperature. In contrast, the MD simulation for **2GCzBPN** demonstrate a negligible temperature-responsive change in the packing (Figure 37).”

Figure S37. (a) Optimized geometry of 2GCzBPN employed for the MD simulation. (b) Radial distribution function calculated as a function of the distance between acceptors (highlighted in orange in a) in the aggregate. The simulation results demonstrate a negligible temperature-responsive change in the packing as the radial distribution function $g(r)$ does not change significantly with temperature.

Comment 3: Considering the broad readers in Nature Communications, an example of actual potential application of 2GCzBPN as thermal sensors should be described.

Our response: Throughout the systematic investigations, we have demonstrated the novelty of this work from four perspectives: emitter design, versatile applications, high-performance temperature sensing, and new mechanism. These have been explained in detail in our response to Reviewer 1. We believe our work is appropriate in terms of novelty, significance, and relevance for publication in *Nat. Commun.*

Given the broad temperature sensing range and outstanding color response of our TADF dendrimer system, we see potential for applications in the following areas:

1. Real-time monitoring for the cold-chain logistics of vaccines and biologics.
2. Environmental monitoring in extreme conditions such as polar regions, deserts, and space.
3. Industrial temperature monitoring.

However, such studies are beyond the scope of this manuscript.

I thank the reviewers for their time in providing feedback on our manuscript.

Sincerely,

Eli Zysman-Colman on behalf of the co-authors

REVIEWERS' COMMENTS

Reviewer #2 (Remarks to the Author):

The authors significantly improved the quality of the manuscript by addressing all the comments of the reviewers and its current version can be published on this journal as it is.

Reviewer #3 (Remarks to the Author):

[Note from the Editor: Reviewer #3 was asked to assess also the response given to reviewer #1 who was not able to look over the revision.]

I have verified that the author's response to my question is appropriate. And also, I considered the authors' replies to Reviewer 1. Reviewer 1 had some critical comments, especially, novelty of this work as the submitted manuscript was originally deposited to ChemRxiv. In the revised manuscript, the authors accepted the reviewer's concerns and responded them with sufficiently persuasive revisions. I feel that the authors' responses are convincing. So I recommend the manuscript to be published in Nature Communications.